# Ectopic Expression of *PtrLBD39* Retarded Primary and Secondary Growth in *Populus trichocarpa*

**DOI:** 10.3390/ijms25042205

**Published:** 2024-02-12

**Authors:** Jing Yu, Boyuan Gao, Danning Li, Shuang Li, Vincent L. Chiang, Wei Li, Chenguang Zhou

**Affiliations:** 1State Key Laboratory of Tree Genetics and Breeding, Northeast Forestry University, Harbin 150040, China; yj74226@163.com (J.Y.); 18231430136@163.com (B.G.); lidanning@nefu.edu.cn (D.L.); shuangli@nefu.edu.cn (S.L.); vchiang@ncsu.edu (V.L.C.); weili2015@nefu.edu.cn (W.L.); 2Forest Biotechnology Group, Department of Forestry and Environmental Resources, North Carolina State University, Raleigh, NC 27695, USA

**Keywords:** PtrLBD39, wood formation, secondary cell wall, primary growth, *Populus trichocarpa*

## Abstract

Primary and secondary growth of trees are needed for increments in plant height and stem diameter, respectively, affecting the production of woody biomass for applications in timber, pulp/paper, and related biomaterials. These two types of growth are believed to be both regulated by distinct transcription factor (TF)-mediated regulatory pathways. Notably, we identified PtrLBD39, a highly stem phloem-specific TF in *Populus trichocarpa* and found that the ectopic expression of *PtrLBD39* in *P. trichocarpa* markedly retarded both primary and secondary growth. In these overexpressing plants, the RNA-seq, ChIP-seq, and weighted gene co-expression network analysis (WGCNA) revealed that PtrLBD39 directly or indirectly regulates TFs governing vascular tissue development, wood formation, hormonal signaling pathways, and enzymes responsible for wood components. This regulation led to growth inhibition, decreased fibrocyte secondary cell wall thickness, and reduced wood production. Therefore, our study indicates that, following ectopic expression in *P. trichocarpa*, PtrLBD39 functions as a repressor influencing both primary and secondary growth.

## 1. Introduction

Trees, vital to the forest ecosystem, serve as valuable renewable energy sources. Tree wood, its primary product, is used for energy combustion, pulping, paper production, and potential lignocellulosic biofuel production [1]. Wood, also known as secondary xylem, develops through the cyclic vascular cambium activity. This process encompasses vascular cambium cell differentiation into secondary xylem mother cells, cell expansion, mass deposition of secondary walls and programmed cell death, and final heartwood formation [2,3]. The secondary xylem includes vessel cells, fibrocytes, and axial parenchyma cells [4], influencing wood properties based on their proportions [5].

Wood formation regulation involves a network of transcription factors (TFs) and secondary cell wall (SCW) component genes [6,7,8,9,10,11,12]. Vascular-related NAC domain (VND) and secondary wall-associated NAC domain (SND) TFs are considered master wood formation regulators [8,10,13,14]. In lower layers, MYB, bHLH, and other gene families regulate wood development in a complex and orderly manner [10,12,15,16]. Additionally, plant hormones play a significant role in various plant growth and development processes [1,17].

The Lateral Organ Boundaries Domain (LBD) family protein, or ASYMMETRIC LEAVES2-LIKE (ASL) protein, constitutes plant-specific TFs featuring conserved Lateral Organ Boundary (LOB) domains [18,19]. It plays a crucial role in leaf, lateral root, inflorescence, and embryo development in plants [20,21,22,23,24]. Additionally, LBDs respond to environmental signals and regulate secondary growth in plants [25,26,27,28]. In poplar, *PtaLBD1*, *PtaLBD4*, *PtaLBD15*, and *PtaLBD18* displayed higher transcript abundance during secondary growth [26]. PtaLBD1 inhibition leads to decreased diameter growth and highly irregular phloem development [26]. Overexpressing *PagLBD3* boosts poplar stem secondary growth and accelerates plastic layer cell differentiation into the phloem, while its dominant inhibition decreases this differentiation [29]. Furthermore, in *Eucalyptus grandis*, LBD37 and LBD29 regulate secondary xylem and phloem development, respectively [27]. Moreover, ectopic *LBD* expression in *Arabidopsis thaliana* induced callus formation [25]. Notably, LBD4 defines the phloem–protocambium boundary shaping vascular bundles in *Arabidopsis* [30]. *Arabidopsis* LBD18 reportedly regulates tracheid element (TE) differentiation during secondary xylem development [31]. Therefore, these collective findings indicate the wide-ranging regulatory function of LBDs in primary and secondary growth across woody and herbaceous plants.

We analyzed the expression of *LBD* genes in stem–cambium, stem–xylem, stem–phloem, shoot, leaf, and root of *Poulus trichocarpa* and found two highly stem–phloem-specific members, *PtrLBD39* and *PtrLBD22*. In this work, we focused on the function of PtrLBD39. While previous studies showed that LBDs exhibit different roles in plant growth, we found that transgenic *P. trichocarpa* plants expressing *PtrLBD39* under a cauliflower mosaic virus 35S promoter resulted in severely stunted phenotypes. Therefore, we hypothesized that ectopic expression of *PtrLBD39* disrupted the regulation of plant growth and development-related genes. In this study, we conducted an analysis encompassing growth, anatomical phenotypes, transcriptomics combined with ChIP-seq [16], and weighted gene co-expression network analysis (WGCNA) of *PtrLBD39* overexpression plants. We found that ectopic overexpression of *PtrLBD39* inhibits both primary and secondary growth in transgenic plants by regulating genes across multiple biological pathways.

## 2. Results

### 2.1. Tissue-Specific Expression of PtrLBD39 in P. trichocarpa

We previously used laser capture microdissection (LCM) to obtain the gene expression levels of different stem vascular tissues (stem–cambium, stem–xylem, and stem–phloem) [32]. The LCM RNA-seq data were normalized with the gene expression levels of leaves, roots, and shoots of *P. trichocarpa* [33], and transcription factors specifically expressed in these six tissues were screened. Among them, we found two highly stem–phloem-specific members, namely *PtrLBD39* and *PtrLBD22* (Figure 1). In this study, we selected PtrLBD39 for functional analysis in transgenic *P. trichocarpa*.

### 2.2. Ectopic Expression of PtrLBD39 in Transgenic P. trichocarpa Resulted in Stunted Phenotypes

Seven independent transgenic lines (*OE-PtrLBD39*-L1~L7) exhibited elevated *PtrLBD39* transcript abundances in xylem (Appendix A). The three lines (L1, L2, L3) demonstrating the highest expression were chosen for subsequent analysis. In comparison to the wild type (WT), overexpressed plants displayed markedly reduced height (Figure 2A,B) and smaller, more curled leaves (Appendix A), indicating that ectopic expression of *PtrLBD39* transgenic significantly inhibited the growth of *P. trichocarpa*. There was no significant growth improvement in transgenic plants over time. *OE-PtrLBD39* significantly increased the internode number while reducing their length compared to WT (Figure 2C and Appendix A), thereby indicating that extreme internode shortening, prompted by *PtrLBD39* overexpression, contributed to the dwarf phenotype. Furthermore, stem basal diameter analysis revealed a significant decrease in *OE-PtrLBD39* compared to WT (Figure 2D).

To investigate the impact of PtrLBD39 on wood formation, we paraffin-sliced and stained the 4th, 6th, 8th, 10th, and 20th stem segments of the greenhouse-grown 4-month-old WT and *OE-PtrLBD39* plants with Safranin O and Fast Green dye for observation (Figure 3). Comparative analysis revealed that *OE-PtrLBD39* exhibited non-lignified fiber cell walls in the xylem and phloem regions, distinctly differing from WT (Figure 3A,B). This demonstrates that *PtrLBD39* overexpression strongly suppresses xylem and phloem fiber cell wall lignification, suggesting its crucial regulatory role in fiber cell development. Additionally, these transgenic plants displayed a significant reduction in single vessel area while the fiber cell area increased compared to the wild type (Figure 3C). Furthermore, *OE-PtrLBD39* exhibited an increased vessel count per unit area and a reduced number of fiber cells (Figure 3D). However, the morphological structure of cambium cells in *OE-PtrLBD39* remained largely unchanged from that of WT (Appendix A).

Since the ectopic expression of *PtrLBD39* affects the fiber cell wall lignification, both scanning electron microscopy (SEM) and transmission electron microscopy (TEM) were used to observe the cell walls of *OE-PtrLBD39* transgenic plants. SEM results demonstrated significant thinning of the fiber cell wall compared to WT, while the vessel cell wall remained largely unchanged (Figure 4A). Moreover, TEM analysis results revealed a significantly thinner fiber cell wall in *OE-PtrLBD39* plants, with the absence of the S3 layer and partial S2 layer of the secondary cell wall (SCW), as compared to the WT (Figure 4B). These results indicated that the ectopic expression of *PtrLBD39* directly affected fibrocyte cell wall development in the vascular tissue of *P. trichocarpa*.

### 2.3. The Ectopic Expression of PtrLBD39 Significantly Affects the Lignin Structure and Polysaccharide Contents

In light of the inhibited fiber cell wall development in transgenic plants, we quantified the three wood cell wall components. While total lignin content in stem wood of *OE-PtrLBD39* transgenics remained unchanged compared to WT (Table 1), there was a significant alteration in lignin composition. This alteration showed a ~14% reduction in S-monomers and ~34% increase in G-monomers, leading to S:G ratios ranging from 0.8 to 1.63 in the transgenics as compared with 2.0 in the wild type (Table 2), thereby aligning with anatomical phenotypes observed in paraffin sections of *OE-PtrLBD39* transgenics (Figure 3A). Furthermore, significant reductions were evident in the total wood glucose (~12% reduction) and xylose (~21% reduction) (Table 1), thus suggesting that cellulose and hemicellulose biosynthesis were strongly inhibited in the presence of a high level of *PtrLBD39* transcripts in xylem. We then examined the effects of *PtrLBD39* overexpression on the transcript levels of downstream genes.

### 2.4. Identification of Downstream Genes of PtrLBD39

We conducted RNA-seq on stem-differentiating xylem (SDX) tissues from *PtrLBD39* ectopic expression transgenic plants (L1, L2, and L3) and wild-type plants. In *OE-PtrLBD39*-L1, L2, and L3, we found 18502, 9823, and 9474 differentially expressed genes (DEGs), comprising 17336, 8126, and 7926 upregulated genes, and 1166, 1697, and 1548 downregulated genes, respectively (FDR < 0.05, |LogFC| > 1; Appendix A). Combining the DEGs from these three lines, we identified a total of 6151 DEGs, comprising 5616 upregulated genes and 535 downregulated genes (Appendix A). To comprehend their function, KEGG functional enrichment analysis revealed enrichment in pathways such as plant hormone signal transduction, MAPK signaling pathway–plant, starch and sucrose metabolism, biosynthesis of secondary metabolites, phenylpropanoid biosynthesis, phenylalanine metabolism, and others (Appendix A).

In *OE-PtrLBD39*, the known monolignol biosynthetic pathway genes and SCW cellulose biosynthetic genes (*PtrCesA4*, *PtrCesA7*, *PtrCesA17*, *PtrCesA8*, and *PtrCesA18*) exhibited significant reduction in transcript levels [34,35] (Figure 5A and Appendix A). Similarly, genes associated with hemicellulose biosynthesis showed a considerable decrease in expression (Figure 5C and Appendix A). These changes in cell wall component genes corresponded to alterations in wood components (Table 1 and Table 2).

After ectopically expressing *PtrLBD39*, we observed stunted plant growth, indicating its influence on primary and secondary growth in trees. This overexpression notably altered the transcription levels of *STM* and *LBD15*, both crucial for plant primary and secondary growth. *PtrLBD39* overexpression upregulated the transcription factor STM, encoding the KNOX1 protein (Appendix A) [36]. Conversely, the expression of *LBD15*, involved in stem apical meristem development and regulation of secondary xylem cell differentiation, decreased (Appendix A) [37,38]. Moreover, several hormone response factors governing tree growth and development were upregulated post-*PtrLBD39* ectopic expression. Notably, cytokinin response factors CRF2, CRF4, and CRF11 showed elevated expression levels. Additionally, *ERF1*, a direct target of EIN3 in the ethylene signaling pathway, exhibited increased expression (Appendix A) [39]. Moreover, the expression levels of *GA2OX6*, *GA2OX8*, *GA3OX1* and *GA20OX2* in gibberellin (GA) metabolism were altered (Appendix A). Significantly, distinct alterations in the expression levels of key TFs associated with secondary growth were observed. For instance, *ANT* was primarily expressed in the vascular cambium, and, along with *APL* genes, was essential in regulating *Arabidopsis thaliana* phloem properties also being upregulated (Appendix A) [1,40]. The differential expression of these TFs, governing primary growth, secondary growth, and hormonal responses in *OE-PtrLBD39*, suggests the potential influence of PtrLBD39 on the wood formation process through the regulation of these TFs.

Upon analyzing ChIP-seq data from *OE-PtrLBD39-FLAG* transgenic plants [16], we identified 162 upregulated DEGs and 11 downregulated DEGs, associate targets of PtrLBD39 (Appendix A). The binding motifs within the promoters of PtrLBD39 target genes were analyzed using STREME [41]. The results showed that four top-ranked motifs in binding peaks of targets’ promoter for TF binding (*p* value < 2.2 × 10^−8^; Appendix A), indicated that PtrLBD39 may regulate the target genes by binding to these potential motifs. Among the TF genes bound by PtrLBD39, HD-ZIP family TFs HB6 and ARF family TFs MP, linked to hormone response, exhibited upregulation (Appendix A) [42,43]. The *STY1* gene, involved in auxin biosynthesis, showed a modest increase in transcription levels [44,45]. Additionally, the expression level of the growth-related *NERD* increased by 2–3 times (Appendix A) [46]. *PtrMYB128* (*MYB103*), a member of the transcriptional network regulating secondary cell wall biosynthesis, was directly downregulated by PtrLBD39 (Appendix A) [47,48,49]. These changes in the expression levels of TFs involved in hormone signaling response and secondary growth confirmed PtrLBD39’s regulation of these genes through promoter binding, thereby regulating tree growth.

### 2.5. WGCNA Reveals Gene Modules Associated with Phenotypes

To identify genes linked to the *OE-PtrLBD39* trait, we performed WGCNA analysis on DEGs in *OE-PtrLBD39* RNA-seq data (Appendix A) and correlated phenotypic data encompassing primary (plant height and stem segment count), secondary growth (stem basal diameter), and wood properties (wood composition and lignin content) (Appendix A). Unexpressed or under-expressed genes were excluded from analysis, with WGCNA detection being performed for genes with an average FPKM expression >1. Utilizing the soft threshold calculation (Appendix A), *β* = 20 was chosen to construct a network. The dynamic shear tree method combined similar expression modules, yielding 10 co-expression modules (differentiated by different colors) comprising gene counts in different modules ranging from 215 to 6405. Each highly correlated gene set corresponds to a branch in the gene dendrogram (Appendix A), with module–phenotype correlations demonstrating significant overlap between genes in the same module (Figure 6). Notably, three modules (yellow, black, and brown) exhibit the strongest correlations with the *OE-PtrLBD39* trait data (*q* < 0.05, |R| > 0.85) (Figure 6), suggesting that these genes primarily impact tree growth and development. Consequently, we further scrutinized these three modules.

### 2.6. Functional and Pathway Enrichment Analysis of the Eigengenes in Modules

There were 771, 107, and 790 DEGs in the yellow module, black module, and brown module, respectively (Appendix A). Correlation analysis showed that the yellow module exhibited positive correlations with plant height, basal diameter, H-type lignin, and glucose content, while being negatively correlated with the internode number. The black module primarily relates to lignin components: S-type lignin content and S/G ratio were significantly negatively correlated, while G-type lignin content showed a positive correlation. Conversely, the brown module displayed an opposing pattern to the yellow module concerning plant growth traits; for instance, plant height and number of basal diameters exhibited significant negative correlations (Figure 6). This suggests a mutual inhibiting effect on tree growth between genes in the yellow and brown modules.

To explore the role of DEGs in the yellow, black, and brown modules, we performed GO functional enrichment analyses on their respective genes (Appendix A). Data analysis revealed significant enrichment of genes in these modules linked to various GO terms encompassing biological processes, cellular components, and molecular functions. Primarily, these genes are associated with metabolic and cellular processes, cell components, as well as binding and catalytic activities (Appendix A). Notably, the molecular function category, particularly the binding term, contains the largest number of genes. Binding is implicated in diverse biological processes such as signal transduction, metabolic pathways, gene regulation, and protein interactions, suggesting its involvement in regulating plant growth and development. MYBs and NACs, specifically known for their role in secondary cell wall synthesis and xylem differentiation, are notably abundant within the ‘binding’ term [8,12,50]. Furthermore, this term encompasses genes associated with hormone signaling pathways, including auxin response factor *ARFs* and *ERFs* involved in ethylene synthesis. These findings strongly suggest that these genes, particularly the *MYB* and *NAC* family genes along with those related to hormone signaling pathways, may contribute to the emergence of the *OE-PtrLBD39* phenotype. They potentially play a crucial role downstream of PtrLBD39 in tree growth and morphogenesis.

### 2.7. WGCNA Reveals the Genes Expression Directly Related to Phenotype in P. trihcocarpa

In the examination of genes regulated by PtrLBD39 within the modules and their impact on phenotypes, the integration and analysis of *OE-PtrLBD39* ChIP-seq data [16] with DEGs in three modules revealed that 27 target genes were regulated by PtrLBD39 in the yellow module. This set includes six TFs (Appendix A). Notably, *PtrMYB128*, (poplar fiber cell wall thickness regulation) and *PtrCAD1* (cinnamyl alcohol dehydrogenase; a monolignol biosynthesis enzyme) were significantly downregulated in *OE-PtrLBD39* (Appendix A) [7,47,48,49,51,52]. Moreover, the TF MP/ARF5, known for mediating Arabidopsis morphogenesis and vascular development, displayed significant upregulation in *OE-PtrLBD39* (Appendix A) [43].

In the black module, PtrLBD39 directly regulates three target genes, notably including the upregulation of the *EOL1* gene associated with Arabidopsis ethylene biosynthesis in *OE-PtrLBD39* (Appendix A) [53]. The brown module reveals PtrLBD39’s direct regulation of 38 target genes, encompassing six TFs. These genes are linked to plant short stem, lateral root development, leaf, and cell morphogenesis [42,54,55,56,57]. For instance, *Arabidopsis thaliana*’s homologous genes—*LRL3*, *HB6*, *WRKY28*, *SRS5*, and *OBP1*—were all upregulated in *OE-PtrLBD39* (Appendix A). These analyses indicate that PtrLBD39-mediated direct regulation of target genes within the modules potentially influenced both the primary and secondary tree growth as well as xylem secondary cell wall development.

## 3. Discussion

As per the transcriptome data of six tissues (stem–cambium, stem–xylem, stem–phloem, shoot, leaf, and root) in *P. trichocarpa*, *PtrLBD39* is highly specific in the stem–phloem (Figure 1). Upon overexpressing *PtrLBD39* in *P. trichocarpa*, its expression in the xylem of different transgenic lines was 1000–7500 times higher (Appendix A). However, its ectopic expression of *PtrLBD39* severely reduced the plant height, leaf size, and stem basal diameter (Figure 2 and Appendix A), thus profoundly impacting the primary and secondary growth. Overexpression of *PtrLBD39* resulted in diverse alterations in cell size and number within xylem fiber and vessel cells (Figure 3); however, it did not impact the stem vascular cambium cells (Appendix A). This suggests that ectopic expression of *PtrLBD39* affected the expansion and proliferation of specific cell types. In the transgenic plants, the secondary cell wall of the xylem vessels resembled that of the wild type, but the xylem fiber cell wall was significantly thinner (Figure 4). Consistent with the phenotype of fiber cell wall, glucose and xylan, the primary components of cellulose and hemicellulose, respectively, were significantly reduced in *OE-PtrLBD39* (Table 1). Lignin in dicotyledonous plants mainly comprises S-type and G-type lignin, which mainly comes from fiber and vessel cells, and usually exists in a ratio of 2:1, respectively [58,59,60,61]. We found that ectopic expression of *PtrLBD39* altered lignin structures in the cells. The lignin alteration reduced lignin S:G ratios by 20–60% due to *PtrLBD39* ectopic expression, explaining the non-lignification of the fiber cell wall but normal lignification of the vessel cell wall (Table 2 and Figure 3). These phenotypes of cell wall and component changes were consistent with the reduction in abundance of cell wall component gene transcripts (Figure 5).

In the transcriptome data of *PtrLBD39* overexpression, we observed upregulation of *STM* and *LBD15*, both known for their roles in regulating primary and secondary plant growth (Appendix A) [62]. STM, a KNOX1 protein class, reportedly inhibited cell differentiation by inducing *IPT7* and A-type *ARR5* expression [36]. *ARK1*, a poplar homolog of *STM*, reportedly altered cell wall biochemistry and lignin composition when overexpressed in plants [62], likely through inhibiting cell differentiation. The upregulation of *STM* in *OE-PtrLBD39* plants likely leads to reduced height, altered cell wall thickness, and wood composition. Conversely, *OE-PtrLBD39* resulted in a decreased transcriptional abundance of *LBD15*. Studies have shown that *AtLBD15* regulates stem apical meristem development in *Arabidopsis* and regulates *WUS* expression [37], potentially inhibiting the primary growth in *PtrLBD39* overexpressed plants. LBD15 also plays a role in secondary cell wall synthesis during xylem cell differentiation [7,38,63]. Increased *LBD15* expression in *Arabidopsis* was found to downregulate *CesA4*, *CesA7*, and *CesA8*, which are vital for mediating secondary cell wall cellulose synthesis, thereby disrupting secondary cell wall formation [38]. Although a decrease in *LBD15* expression would theoretically upregulate these cellulose synthase genes, the cellulose synthase gene and secondary cell wall formation are regulated by multiple factors.

Plant hormones regulate plant development. Ethylene influences secondary stem growth by stimulating cambium activity, xylem development, and fibre and vessel formation. Ethylene binds to ethylene receptors (ETRs) at the endoplasmic reticulum, initiating downstream signaling pathways involving phosphorylation, ubiquitination, and translocation, activating a set of TFs, namely ETHYLENE INSENSITIVE 3/ETHYLENE INSENSITIVE3-LIKE1 (EIN3/EIL1) and Ethylene Response Factor (ERF) TFs [64]. *ERF1*, a direct target of EIN3, is ethylene-sensitive [39]. Therefore, the upregulated *ERF1* expression likely contributes to the unique phenotype of transgenic *PtrLBD39*. Cytokinin comprises a class of N6-substituted adenine derivatives that can actively regulate various cellular and developmental processes in plants [65]. During the cell proliferation phase, cytokinin is necessary to maintain cell proliferation and is a key factor in mediating leaf cells from proliferation to expansion by blocking the cell transition to expansion and the initiation of photosynthesis [66]. Cytokinin Response Factors (CRFs), such as CRF2, CRF4, and CRF11, members of the APETALA2 TF family, are upregulated by cytokinin transcription. They regulate gene expression involved in cytokinin biosynthesis (*IPT* and *LOG*) and degradation (cytokinin oxidase/dehydrogenase, *CKX*) and positively regulate type-A *ARRs* [36]. It has also been shown that CKG, a cytokinin response factor, mediates CK-dependent regulation of Arabidopsis cell expansion and cell cycle progression. Overexpression of *CKG* can increase cell size and promote cell cycle progression. The mutant *CKG* in Arabidopsis shows smaller cotyledon [67]. The elevated expression levels of *CRF2*, *CRF4*, and *CRF11* in *OE-PtrLBD39* may be one of the main reasons for the alterations in the number of xylem fiber and vessel cells (Figure 3). Bioactive gibberellin also plays a variety of roles in plant development [68,69,70]. Gibberellin can promote plant growth, including promoting plant cell division and cell elongation [71]. In higher plants, the content of endogenous active gibberellin is balanced by its biosynthesis and catabolism, which determine the regulatory behavior of gibberellin during plant tissue development. In our transcriptome data, the expression level of seven genes related to GA biosynthesis (*GA3OX* and *GA20OX*) and GA catabolism (*GA2OX*) were altered (Appendix A), which may cause changes in the content and distribution of GA in transgenic plants and is responsible for affecting plant height and leaf size of transgenic plants.

An increasing number of studies have proven that crosstalk among different plant hormones has an effect on plant growth [72,73,74,75]. Auxin, cytokinin, ethylene, and GA affect the activity of the stem vascular cambium [74,75,76]. Changes in the content and distribution of these plant hormones in different types of cells are responsible for cell division and differentiation, as well as fiber and vessel SCW formation, including SCW deposition, wall thickening, and cellulose biosynthesis [73,77]. In this study, the ectopic expression of *PtrLBD39* resulted in the alteration in expression levels of numerous genes related to plant hormone metabolism and regulation, stem apical tissue and secondary cell wall composition (Appendix A, Appendix A), which may have changed the hormone homeostasis in transgenic plants, thus seriously affecting the primary and secondary growth of the plants.

Following an integrated RNA-seq and ChIP-seq analysis [16], we identified 162 genes upregulated and bound to PtrLBD39, among which 28 are TFs. Notably, these TFs encompass MP and NERD and are crucial in primary growth processes. MONOPTEROS (MP) acts as auxin response TFs which are vital for root formation and vascular development [43]. NERD interacts with the exocyst, impacting root growth and cell expansion. The exocyst, a conserved octamer protein complex, mediates plasma membrane secretion and regulates developmental processes like root meristem size, cell elongation, and tip growth. NERD1 indirectly interacts with the exocyst, potentially influencing cell wall polysaccharides and developmental traits [46]. The variations in plant height, cell wall thickness, and wood composition due to *PtrLBD39* ectopic expression likely result from changes in *MP* and *NERD* transcription levels. PtrLBD39 also directly upregulates other TFs like HB6 and STY1, which are involved in hormone regulation. Studies have shown that HB6 (ARABIDOPSIS THALIANA HOMEBOX 6) regulates leaf edge development. HB6 participates in cell division and organ differentiation while negatively regulating ABA signaling during abiotic stress in vitro [42]. STY1, a transcriptional activator of the auxin biosynthesis gene, and its related genes promote the style’s development by regulating auxin homeostasis and the pistil’s apical pattern [44,45]. The alteration in leaf morphology due to *PtrLBD39* ectopic expression could be influenced by these two genes. Moreover, PtrLBD39 was found to downregulate 11 genes. Notably, the TF PtrMYB128 acts as a master switch in *P. trichocarpa*’s secondary wall biosynthesis, previously identified as a member of the transcriptional network regulating secondary wall biosynthesis in Arabidopsis xylem tissue. It is a direct transcriptional target of SECONDARY WALL ASSOCIATED NAC DOMAIN PROTEIN 1 (SND1), which has been proposed to influence cellulose biosynthesis [47,48,49]. Consequently, *PtrMYB128* downregulation might account for reduced secondary cell wall thickness in transgenic plants.

To explore the association more accurately between abnormal phenotypes and DEGs in plants overexpressing *PtrLBD39*, we conducted a WGCNA. Out of the 10 co-expression modules, we obtained three target modules for in-depth analysis. Notably, genes within the yellow and brown modules exhibited antagonistic roles in primary plant growth regulation, while the yellow, black, and brown modules collectively regulated both primary and secondary tree growth. It is noteworthy that the downregulation of *MYB85*, a key regulator of phenylalanine and lignin biosynthesis during secondary cell wall formation [78], and its dominant inhibition significantly reduces secondary wall thickening in fiber cells [79], is downregulated in the yellow module (Appendix A), may result in the defective phenotype of *OE-PtrLBD39*. KNAT1 is an important apical meristem maintenance regulator, with dual effects of promoting cell proliferation and inhibiting differentiation [80]. It was upregulated in the brown module (Appendix A), which may have resulted in reduced plant height. Furthermore, the upregulated expression of *NTL9*, a member of the NAC TF family known to negatively regulate vascular cambium development during secondary growth of *Arabidopsis* stems [81], inhibited secondary growth in *OE-PtrLBD39* plants (Appendix A).

In conclusion, *PtrLBD39* ectopic expression impacts the expression of gene sets crucial for primary and secondary plant growth, leading to reduced plant height and radial growth. This suggests that PtrLBD39 can regulate the growth of trees as an inhibiting factor.

## 4. Materials and Methods

### 4.1. Plant Materials and Growth Conditions

All experiments were performed with *Populus trichocarpa* genotype Nisqually-1. Wild-type and transgenic plants were grown in a greenhouse as described previously [82]. Four-month-old stems of *P. trichocarpa* plants were used for RNA extraction, growth index measurement, histological analysis, and scanning electron microscopy. Six-month-old plants were used for wood composition analysis and transmission electron microscopy.

### 4.2. Generation of Gene Overexpression Transgenic P. trichocarpa

The coding sequence of PtrLBD39 (Potri.005G097800) was amplified from *P. trichocarpa* plants and cloned into the pBI121 vector under the control of the CaMV 35S promoter for generating overexpression construct. The plasmid was introduced into Agrobacterium tumefaciens strain GV3101 for *P. trichocarpa* transformation as described in detail previously [83]. The expression level of *PtrLBD39* gene in the xylem of transgenic plants was detected by RT-qPCR. Primers are listed in Appendix A.

### 4.3. RT-qPCR

Total RNA was isolated from the SDX tissues of *P. trichocarpa* by using the Qiagen RNeasy Mini Kit. One microgram of total RNA was reverse-transcribed to cDNA using the PrimeScript RT Reagent Kit with gDNA Eraser (Takara, Dalian, China). All RT-qPCR reactions were performed on an Agilent Mx3000P QPCR System with FastStart Universal SYBR Green Master (Roche, Basel, Switzerland). PtrActin was used as a housekeeping gene to calculate the relative expression level in wild-type and transgenic plants. Three biological replicates were used for each RT-qPCR experiment. The relative expression levels were calculated by the 2^−ΔΔCt^ method [84]. The primers are listed in Appendix A.

### 4.4. Histochemical and Histological Analysis

Four-month-old wild-type and transgenic plants stem internodes were cut into 2 mm fragments and fixed with formalin–acetic acid–alcohol liquid (FAA, Washington, DC, USA) solution. The fixed stem segments were dehydrated in a graded ethanol series (50%, 70%, 85% and 95%; *v*/*v*) at 4 °C and incubated in ethanol: xylene solution (1:0, 1:1, 0:1, *v*/*v*) at room temperature. Then, the segments were transferred into 3:1 (*v*/*v*) xylene: paraffin solution at 42 °C overnight and immersed in 100% paraffin for 3~4 days. The embedded stem segments were cut into 10-μm sections using a rotary microtome (Leica RM2245) and stained with 5% Safranin O, 1.25% Fast Green and 0.1% Toluidine Blue, respectively. Stem section micrographs were captured by digital microscope and scanner M8 (Precipoint, Garching, Germany).

### 4.5. Scanning Electron Micrograph Analysis

Fresh stem segments of the 20th internode of *OE-PtrLBD39* transgenic and wild-type plants were collected and coated with gold at 10 mA for 60 s. The samples were imaged using a scanning electron microscope JCM-5000 NeoScope (NIKON, Tokyo, Japan).

### 4.6. Transmission Electron Microscopy

Samples were taken from the 20th stem segments of the wild-type and *OE-PtrLBD39* transgenic plants (6-month-old) and cut into samples of 3 mm length and 1 mm width, then fixed in a fixative solution containing 2.5% glutaraldehyde (stored at 4 °C) by vacuum osmosis. The samples were pumped using a vacuum pump for 30 min to two hours. They were then rinsed with 0.1 M phosphate buffer (pH 6.8), fixed with 1% osmic acid fixing solution, and then rinsed with 0.1 M phosphate buffer (pH 6.8). After multi-stage ethanol dehydration at 4 °C, the samples were put into 100% acetone. Soaked with acetone and embedding solution at room temperature and embedding the stem segments in the next day. After seven days of polymerization, the samples were cut into 50~60 nm sections with an ultra-thin microtome, and then stained with uranium-lead citrate double. Finally, transmission electron microscopy was used to observe and take the pictures.

### 4.7. Wood Chemistry

Stem segments of six-month-old *OE-PtrLBD39* transgenic and wild-type plants were extracted with 90% (*v*/*v*) acetone for 48 h and then transferred into 100% (*v*/*v*) acetone for 14 days with fresh acetone replaced every two days. The air-dried wood was used to quantify the wood composition (acid-insoluble lignin, acid-soluble lignin, and sugars) and lignin composition (S-lignin, G-lignin, and H-lignin) following established procedures with three replicates for each assay [52,85].

### 4.8. RNA-Seq and Data Analysis

RNA-seq was carried out with total RNA of SDX tissues isolated from wild-type and *OE-PtrLBD39* transgenic plants. The RNA-seq libraries of overexpression transgenics (*OE-PtrLBD39*-L1, L2 and L3) and wild-type plants were generated with three biological replicates per sample using the NEBNext Ultra II RNA library preparation kit. A total of 12 libraries were sequenced using an Illumina HiSeq4000 to generate paired-end reads. For all libraries, average read lengths of 150 base pairs (bp) were generated by sequencing. After removing the library index sequences from each read, the clean reads were mapped to the *P. trichocarpa* genome v4.0 (https://phytozome-next.jgi.doe.gov/, accessed on 14 January 2021) using HISAT2 [86]. The raw counts were determined and normalized following the established analysis pipeline. DEGs were characterized by FDR < 0.05 and |LogFC| > 1 by using DESeq2 [87].

### 4.9. Motif Analysis

We used the website https://biasaway.uio.no (accessed on 24 January 2024) to generate a background set of sequences, and then used https://biasaway.uio.no/biasaway/g (accessed on accessed on 24 January 2024) to optimize the G/C content. A custom background option was selected through the STREME tool (https://meme-suite.org/meme/tools/streme, accessed on 26 January 2024) [41]. A genomic option was applied to search for de novo motifs.

### 4.10. Weighted Gene Co-Expression Network Analysis

Weighted Gene Co-Expression Network Analysis (WGCNA) was performed using the WGCNA R software package (v1.7.1) [88] inand was detected for FPKM with average expression values greater than 1. The *p*-values of the WGCNA results were identified using the multiple corrections tests (Benjamini–Hochberg FDR). In WGCNA analysis, the weighted value of correlation coefficient is used, that is, the gene correlation coefficient is taken to the NTH power, so that the connections between genes in the network follow the scale-free network distribution. First, the soft threshold was determined. The ‘pickSoftThreshold’ algorithm was used for soft threshold calculation [88]. According to the correlation coefficient between genes, a hierarchical clustering tree was constructed to cluster genes with similar patterns into one module [88]. Combined with the correlation between modules and samples, the correlation between modules and phenotypes was analyzed. The weighted network was visualized, phenotypic data were added, and correlations between modules and phenotypes were analyzed.

### 4.11. Statistical Analysis

Two-tailed Student’s *t*-tests, ANOVA, and Duncan’s new multiple range test (MRT) were carried out for statistical analysis to determine data significance. Significance levels were defined as * *p* < 0.05, ** *p* < 0.01, or different lowercase letters.

## Figures and Tables

**Figure 1 ijms-25-02205-f001:**
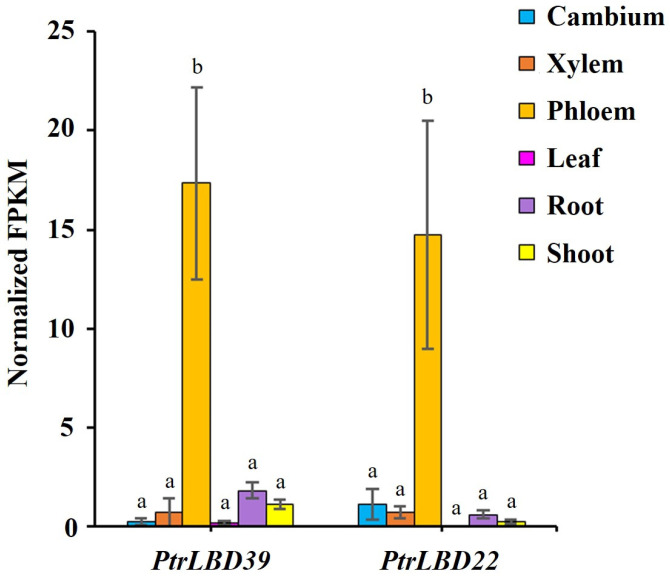
Transcript abundance of *PtrLBD39* in different tissues of *P. trichocarpa*. Normalized FPKM indicates normalized transcript abundances expressed as Fragments Per Kilobase of exon model per Million mapped fragments. The differences analysis was conducted using the IBM SPSS Statistics v27.0 software with Duncan’s new multiple range test (MRT); error bars indicate the SE (standard error) of three biological replicates from independent pools, and bars with different lowercase letters were found to be significantly different (*p* < 0.05).

**Figure 2 ijms-25-02205-f002:**
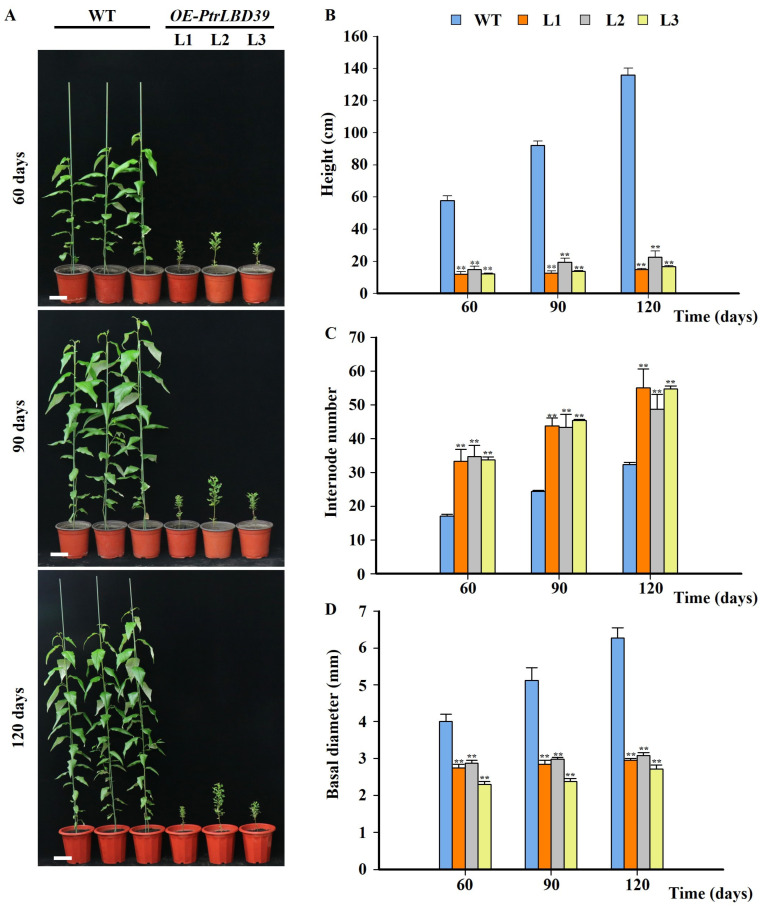
Effects of overexpressing *PtrLBD39* on *P. trichocarpa* growth. (**A**) Phenotypic observations of *P. trichocarpa* wild-type and *PtrLBD39* overexpression plants (*OE-PtrLBD39*-L1, L2, L3) at 60, 90 and 120 days (Bars = 10 cm). (**B**–**D**) Growth indicators determination of wild-type and *PtrLBD39* overexpression plants at different growth periods. Error bars represent SE values of three independent experiments, and asterisks indicate significant differences between the transgenics and wild-type plants found using Student’s *t*-test (** *p* < 0.01).

**Figure 3 ijms-25-02205-f003:**
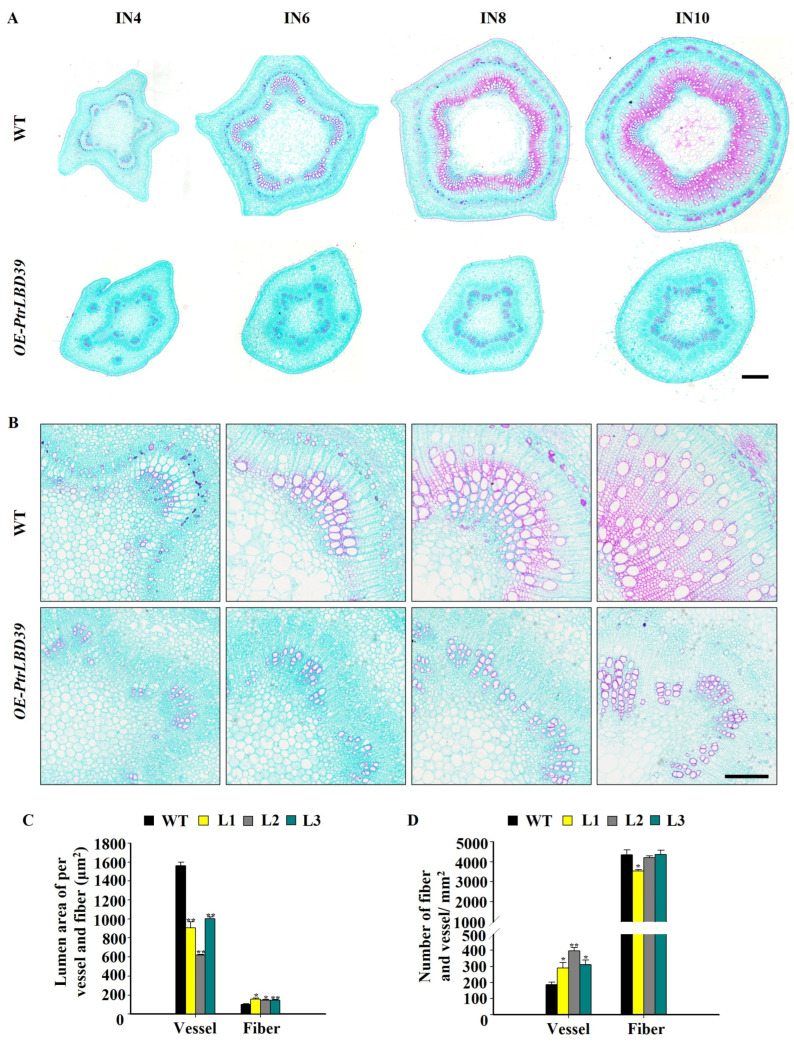
Overexpression of *PtrLBD39* affects the number, lignification of fiber cells and the number and lumen area of vessels. (**A**,**B**) Stem cross sections and magnified images of 4th, 6th, 8th, 10th, and 20th stem segments of 4-month-old *OE-PtrLBD39*-L1 and wild-type plants (The cross sections were stained with Safranin O and Fast Green. (**A**) Bars = 500 μm. (**B**) Bars = 200 μm. IN: internode). (**C**) Statistics analysis of the mean lumen area of individual fiber and vessel cell (mm^2^) within the 20th internode. (**D**) Statistics analysis of the number of fiber and vessel cells per cross-sectional area (mm^2^) within the 20th internode. Error bars represent SE values of three independent replicates with at least 150 vessel cells and 3000 fiber cells for each genotype in each replicate, and asterisks indicate significant differences between the transgenics and wild-type plants found using Student’s *t*-test (* *p* < 0.05, ** *p* < 0.01).

**Figure 4 ijms-25-02205-f004:**
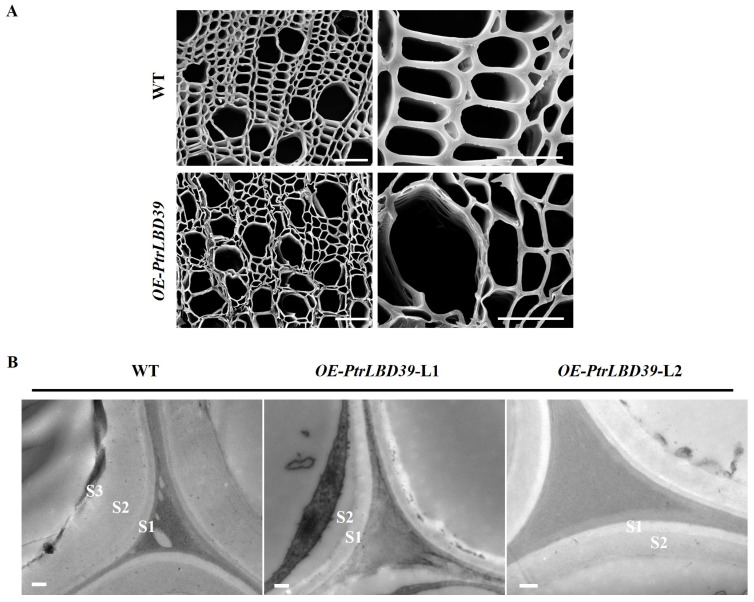
Overexpression of *PtrLBD39* affects the cell wall thickness of fiber cells. (**A**) SEM of WT and *OE-PtrLBD39*-L1 with the 20th internode imaged at ×500 (**left**) and ×2000 (**right**) magnification (Bars = 20 µm). (**B**) TEM images from the 20th xylem fibres in 6-month-old wild-type (WT), *OE-PtrLBD39*-L1 and *OE-PtrLBD39*-L2. S, S-layer of SCW with S1, S2 and S3 (Bar = 500 nm).

**Figure 5 ijms-25-02205-f005:**
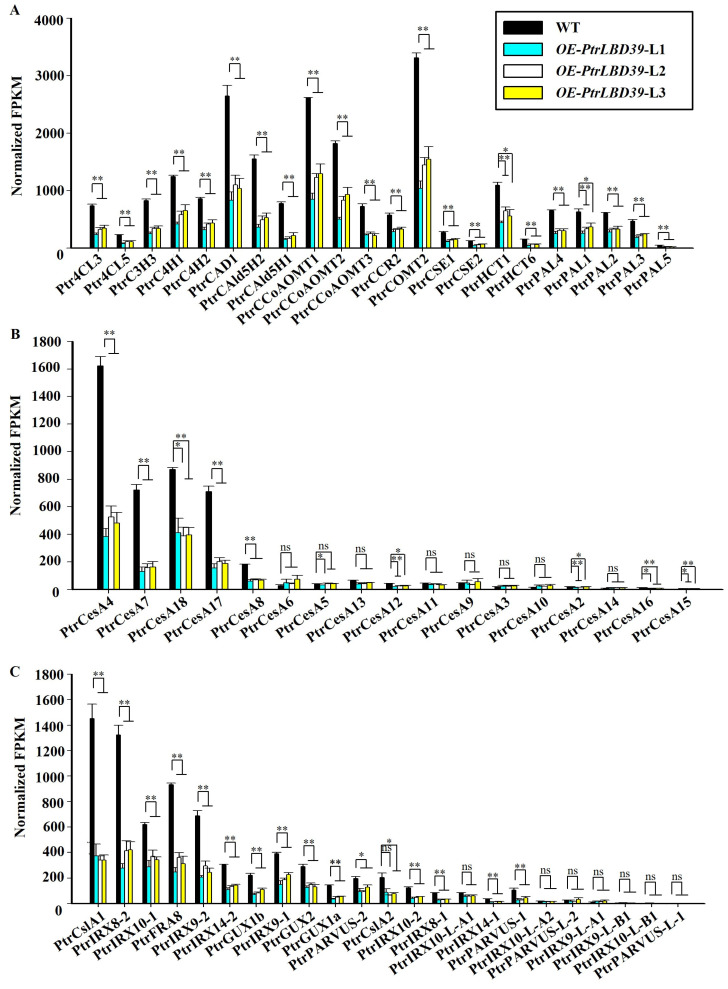
RNA-seq analysis of the transcript abundances of cell wall component genes. (**A**) Monolignol genes. (**B**) Cellulose genes. (**C**) Hemicellulose genes. Normalized FPKM indicates normalized transcript abundances expressed as Fragments Per Kilobase of exon model per Million mapped fragments. Error bars indicate the SE of three biological replicates from independent pools. * *p* < 0.05, ** *p* < 0.01, ns, not significant.

**Figure 6 ijms-25-02205-f006:**
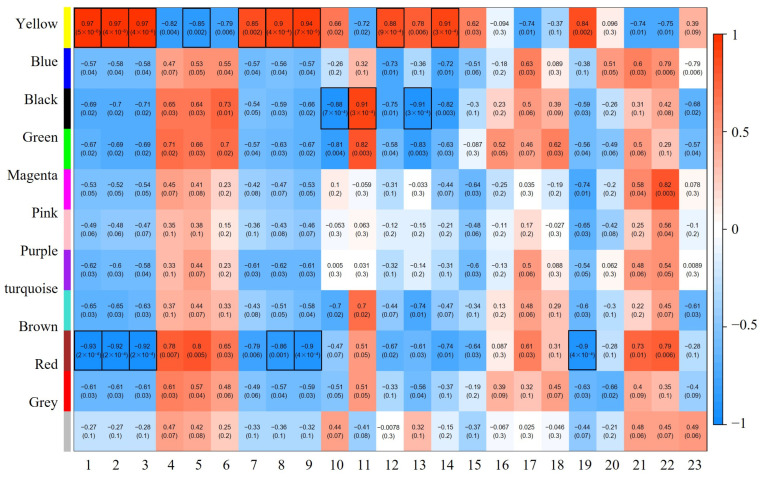
Co-expression, the correlation between gene modules and traits. *X*-axis represents the 23 traits (1–3: plant height at different growth stages, 4–6: internode number at different growth stages, 7–9: basal diameter at different growth stages, 10: S-type lignin content, 11: G-lignin content, 12: H-lignin content, 13: S/G ratio, 14: glucose, 15: xylose, 16: acid insoluble lignin, 17: acid soluble lignin, 18: total lignin, 19: lumen area of individual vessel cell, 20: lumen area of vessel cell per area, 21: lumen area of fiber cell per area, 22: number of fiber cell per area, 23: number of vessel cell per area successively); *Y*-axis represents the co-expression modules (Genes with similar expression patterns are grouped into one module. One color represents one module of genes). The table is color-coded to represent correlation strength according to the color legend. The intensity and direction of each correlation are indicated on the right side of the heatmap (red, positively correlated; blue, negatively correlated). Numbers on the table represent the correlation coefficient between the corresponding module and the trait, with *q*-values listed below the correlations in parentheses. Modules within boxes with black lines: *q* < 0.05, |R| > 0.85.

**Table 1 ijms-25-02205-t001:** Wood composition of *OE-PtrLBD39* transgenic and wild-type *P. trichocarpa*.

Wood Composition	WT	L1	L2	L3
Glucose	54.00 ± 0.34	48.16 ± 0.56 **	47.66 ± 0.62 **	47.5 ± 0.8 **
Xylose	14.84 ± 0.72	11.69 ± 0.37 *	11.66 ± 0.28 *	11.72 ± 0.19 *
Total carbohydrate	72.51 ± 1.39	63.09 ± 1.30 *	61.30 ± 0.52 **	61.39 ± 0.88 **
Acid insoluble lignin	18.90 ± 0.66	19.33 ± 0.09	18.67 ± 0.44	19.02 ± 0.31
Acid soluble lignin	2.21 ± 0.06	3.01 ± 0.15 *	2.52 ± 0.10	3.29 ± 0.09 **
Total lignin	21.12 ± 0.66	22.34 ± 0.24	21.19 ± 0.54	22.30 ± 0.26

Six-month-old plants were tested. Three biological replicates from independent pools of *OE-PtrLBD39* and wild-type (WT) stems were carried out. Data are the means of three independent assays with SE values. Asterisks indicate significant differences between *OE-PtrLBD39* transgenics and wild-type plants by Student’s *t*-test and ANOVA (* *p* < 0.05, ** *p* < 0.01). Units are g per 100 g of dry extractive-free wood.

**Table 2 ijms-25-02205-t002:** Lignin composition of *OE-PtrLBD39* transgenic and wild-type *P. trichocarpa*.

Lignin Composition	WT	L1	L2	L3
S-Lignin	63.7 ± 0.5%	45.5 ± 0.7% **	59.7 ± 0.1% **	59.46 ± 0.6% **
G-Lignin	31.2 ± 0.2%	51.2 ± 0.8% **	36.5 ± 0.3% **	37.4 ± 0.4% **
H-Lignin	5.0 ± 0.3%	3.2 ± 0.3% *	3.8 ± 0.1% *	3.1 ± 0.2% **
S/G ratio	2.0 ± 0.03	0.8 ± 0.02 **	1.63 ± 0.01 **	1.59 ± 0.03 **

Six-month-old plants were tested. Three biological replicates from independent pools of *OE-PtrLBD39* and wild-type (WT) stems were carried out. Data are the means of three independent assays with SE values. Asterisks indicate significant differences between *OE-PtrLBD39* transgenics and wild-type plants by Student’s *t*-test (* *p* < 0.05, ** *p* < 0.01). S, S subunits; G, G subunits; H, H subunits. Values indicate the percentage weight of total lignin.

## Data Availability

The RNA-seq data have been deposited in the National Center for Biotechnology Information Sequence Read Archive as RNA-seq data of *OE-PtrLBD39*, under accession number PRJNA1044722. The ChIP-seq data has been deposited in the National Center for Biotechnology Information Sequence Read Archive as ChIP-seq data of *PtrLBD39* fuse FLAG overexpression plants, under accession number PRJNA706148 [18]. Sequence data from this article can be found in *Populus trichocarpa* genome v4.0 (Phytozome).

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
