# Peer review of "Ectopic Expression of PtrLBD39 Retarded Primary and Secondary Growth in Populus trichocarpa"

_ijms, 2024, doi:10.3390/ijms25042205_

Round 1

Reviewer 1 Report

Comments and Suggestions for Authors

The Manuscript entitled “Ectopic expression of PtrLBD39 retarded primary and secondary growth in Populus trichocarpa” by Yu et al., investigated the effect and possible regulatory network of overexpression of PtrLBD39 in poplar.  Anatomical analysis revealed PtrLBD39's impact on plant growth, xylem development, and modifications in secondary cell wall structure. Additionally, RNA-seq and co-expression analyses identified potential downstream genes responsible for growth inhibition and alterations in cell wall composition. However, it's important to note that the biological functions of these genes cannot be defined through ectopic studies alone. Here are some comments:

Given that this study primarily focused on ectopic expression and that knockout (KO) of PtrLBD39 did not affect plant growth (Yu et al., 2022), it may be advisable for the authors to refrain from conclusively stating "PtrLBD39 as a repressor affecting both primary and secondary growth in P. trichocarpa" in the Abstract.

In addition to leaf images, including data related to leaf size and shape could enhance the analysis. Specifically, it would be beneficial to explore whether OE-PtrLBD39 influences cell expansion, cell proliferation, or both.

While the study addressed the impact of auxin and ethylene, it might be valuable to consider the effects of gibberellin on plant growth. Were any GA biosynthesis or signaling pathway related genes identified through RNA-seq data? Also, could the dwarf phenotype observed in OE-PtrLBD39 be potentially rescued by GA application?

Was there any observation regarding changes in cellulose and semi-cellulose content in OE-PtrLBD39 plants?

It would be insightful to discuss the relationship between PtrLBD39-regulated hormone responses, plant growth inhibition, and lignin composition changes.

Comments on the Quality of English Language

Line 171, please spell out SDX.

Reviewer 2 Report

Comments and Suggestions for Authors

The manuscript is well written, I can advice modern approaches to underline the importance of the results.

66-68

…In this study, we conducted an analysis encompassing growth, anatomical phenotypes, transcriptomics combined with ChIP-seq, and weighted gene co-expression network analysis (WGCNA) of PtrLBD39 overexpression plants.

This is slightly misleading, since your study analyzed but not performed a ChIP-seq experiment

Figure 2, mark axes Y in B-D panels as “Time (days)”, and labels as 60, 90, 120

Though the style “Time, days” is more common.

SE values – explain this

172 - 174

In OE-PtrLBD39-L1, L2, and L3, we found 18502, 9823, and 9474 differentially expressed genes (DEGs), comprising 17336, 8126, and 7926 upregulated genes, and 1166, 1697, and 1548 downregulated genes, respectively

These numbers seem too high, did you apply the fold change criterion to define up/down DEG? Since

453

…DEGs were characterized by FDR < 0.05 by using DESeq2 [77].

But up/down regulated genes required certain threshold for the fold change,

Jung, K., Friede, T. & Beißbarth, T. Reporting FDR analogous confidence intervals for the log fold change of differentially expressed genes. BMC Bioinformatics 12, 288 (2011). https://doi.org/10.1186/1471-2105-12-288

Does Figure S4 means that folds are ignored?

Table S3.

Correct the Strand column or delete it (the ‘-‘ strand is absent), as concern TF families, I would recommend to apply TF classes instead, a new classification of plant TFs derived from mammalian ones is more reasonable, see TFClass http://www.edgar-wingender.de/huTF_classification.html and Plant-TFClass,

Blanc-Mathieu, R., Dumas, R., Turchi, L., Lucas, J., & Parcy, F. (2023). Plant-TFClass: a structural classification for plant transcription factors. Trends in plant science, S1360-1385(23)00227-3. Advance online publication. https://doi.org/10.1016/j.tplants.2023.06.023

As concern

210-212

…Upon analyzing ChIP-seq data from OE-PtrLBD39-FLAG transgenic plants [18], we identified 162 up-regulated DEGs and 11 down-regulated DEGs, direct targets of PtrLBD39 (Table S3).

Ref. 18

Yu, J.; Zhou, C.G.; Li, D.N.; Li, S.; Lin, Y.C.J.; et al. A PtrLBD39-mediated transcriptional network regulates tension wood formation in Populus trichocarpa. Plant Commun. 2022, 3, 100250.

I suspect that de novo motif search in this study was not done quite properly (motif length 6 bp, DREME (Discriminative Regular Expression Motif Elicitation; Bailey, 2011) is too old tool). I recommend apply for de novo motif search the Homer tool http://homer.ucsd.edu/homer/motif/ with genomic background sequences or (it is even better) the STREME tool https://meme-suite.org/meme/tools/streme possessing the web interface, use the custom background option and apply genomic option. STREME from the same team of Bailey et al (2021). Background set of sequences can be generated here https://biasaway.uio.no/ , G/C content optimized https://biasaway.uio.no/biasaway/g/ Unfortunately, this tool allows only A. thaliana genome, i.e. you generate G/C content matched background sequences from A. thaliana genome for the foreground set of your LBD39 peaks. After that, deduced motif allow to deduce direct targets of LBD39 among your genes, use https://meme-suite.org/meme/tools/fimo if you have not your own PWM scanner.

212

Among the TFs bound by PtrLBD39, HD-ZIP family TFs HB6 and ARF family TFs MP

->

Among the TF genes bound by PtrLBD39, HD-ZIP family TFs HB6 and ARF family TFs MP

272

In the examination of genes directly regulated by PtrLBD39 within the modules and

->

In the examination of genes regulated by PtrLBD39 within the modules and

ChIP-seq provides a mix of direct and indirect targets, latter mean a protein intermediate. Since the inherent inability of this technology to distinguish direct and indirect TF-DNA interactions, a substantial portion of ChIP-seq peaks represents the binding sites of partner intermediary TFs and does not represent those for target TFs (Karimzadeh and Hoffman, 2022).

Karimzadeh M, Hoffman MM. Virtual ChIP-seq: predicting transcription factor binding by learning from the transcriptome. Genome Biol. 2022;23:126. https://doi.org/10.1186/s13059-022-02690-2

To use the term ‘direct’ you should apply the LBD39 motif and find potential LBD39 binding sites, see above.

Also,

The phrase in the Abstract

…In these overexpressing plants, the RNA-seq, ChIP-seq, weighted gene co-expression network analysis (WGCNA) revealed that PtrLBD39 directly or indirectly regulates TFs governing vascular tissue development, wood formation, hormonal signaling pathways, and enzymes responsible for wood components.

Can be more concise if you apply the de novo LBD 39 motif.

Similarly, one of the main conclusions of the study, also will be more rigorous

342

Following an integrated RNA-seq and ChIP-seq analysis [18], we identified 162 genes up-regulated and bound to PtrLBD39, among which 28 are TFs.

 What software package is used for WGCENA ?

454

4.9. Weighted Gene Co-Expression Network Analysis

This piece of text implies that there are some specific parameters that should be explained.

227-231

Utilizing the soft threshold calculation (Figure S5), β = 20 was chosen to construct a network. The dynamic shear tree method combined similar expression modules, yielding 10 co-expression modules (differentiated by different colors) comprising gene counts in different modules ranging from 215 to 6405 (Table S5).

If there is no reference, methods section should explain algorithm concerning Figure S5.

Figure S6. Ref. for the tree cutting algorithm ?

Figure S7. Correlation between modules and phenotypes (groups). ->

Figure S7. Correlations between modules and phenotypes (groups).

Figure 6 and Figure S7.

Pearson, Spearman, Kendall ? Why all coefficients are shown? Only the significant one should be represented. The multiple correction should be applied too, scale for significance should be –log10(p-value)m bur not corr. coef.

Figure 6

the level of correlation -> significance of correlation

Meaning of neither Axis X (colors) no Axis Y (traits) is explained properly.

Comments on the Quality of English Language

good

Round 2

Reviewer 1 Report

Comments and Suggestions for Authors

The authors have addressed my questions or given reasonable explanations. I endorse the publication of this manuscript in IJMS.

Author Response

Thank you very much for taking the time to review the revised version of our manuscript, and we appreciate your recognition of our work.

Reviewer 2 Report

Comments and Suggestions for Authors

…Comments 3: SE values – explain this

Response 3: Revised as suggested. SE values represent values of standard error of the

experiments. We have explained SE values in figure legends in this revision (line 85, 103,

122, 155, 161, 185, Figure S1, S2, S3)….

Any abbreviation should be explained only once at first occurrence

Comments 5: Table S3. Correct the Strand column or delete it (the ‘-‘ strand is absent), as concern TF families, I would recommend to apply TF classes instead, a new classification of plant TFs derived from mammalian ones is more reasonable, see TFClass http://www.edgarwingender.de/huTF_classification.html and Plant-TFClass,

Blanc-Mathieu, R., Dumas, R., Turchi, L., Lucas, J., & Parcy, F. (2023). Plant-TFClass: a structural classification for plant transcription factors. Trends in plant science, S1360-1385(23)00227-3. Advance online publication. https://doi.org/10.1016/j.tplants.2023.06.023

Response 5: Revised as suggested (Table S1, S3, S7, S8).

You did not correct your text, see the right class names here https://hal.science/hal-04212079/file/PLANT%20TF%20CLASS_forHAL.pdf Have you read this PlantTFClass paper that I mentioned previously? Correct Table S3 too. Check the manuscript for the word ‘family’.

207

… Upon analyzing ChIP-seq data from OE-PtrLBD39-FLAG transgenic plants [1816], we 207 identified 162 up-regulated DEGs and 11 down-regulated DEGs, associate direct targets 208 of PtrLBD39 (Table S3)…

Overall, you have 17336 genes as up DEG. How many of all ~38.000 genes were mapped to ChIP-seq peaks? Let it be NNN genes. How many peaks you took in analysis? I suspect that 162 out of 17336 is too low portion compared to NNN out of 38000. Explain this or correct your analysis. You can use 2x2 table test to deduce an enrichment (Fisher exact test)

210

STREME is the tool name, I propose do not spell out the abbreviation

I wrote previously,

… 172 - 174. In OE-PtrLBD39-L1, L2, and L3, we found 18502, 9823, and 9474

differentially expressed genes (DEGs), comprising 17336, 8126, and 7926 upregulated genes, and

1166, 1697, and 1548 downregulated genes, respectively.

These numbers seem too high…

Try to explain why according to Table S1 you found about 50% (17336) of all genes (34873) as up-regulated. Random expectation brings about 5%. Any plant has many specific genes, so it seems doubtful that about a half of all genes are up-regulated in your specific tissues and stage.

Overall, you result would be correct if you either correct possible errors in calculation, or apply strongest thresholds for FDR & Fold to select the most important DEGs. In a half of all genes anyone can find anything, but is not a correct analysis.

211

… The results showed that the top 15 motifs were enriched on binding peaks of 136-173 genes (P value < 0.0001; Table S5),…

This phrase is awful. Why this range 136-172 is needed? What can it prove? (I thing nothing). You can find enriched motif in any random set of sequences. If you present the enriched motifs, they should be interpreted as binding sites for certain TFs. If you wrote about enrichment of motifs in promoters of DEG, you should compare the number of predicted genes in non-DEG, and provide p-values, e.g. Fisher exact test

What is aim to provide Sites & Gene ID in Table S4? It  is looks like a garbage.

493-498, 4.9. de novo motif search.

What sequences were used in de novo motif search? Why output motifs were not tested as matching to known ones, e.g.

A random paper for ChIP-seq analysis, https://journals.plos.org/plosone/article?id=10.1371/journal.pone.0089695

You see Logo on Fig. 1, but not only consensuses. For each motif a potential TF with similar motif is shown, in STREME this is done by a tool TomTom almost automatically, so please replace your Table S5, there I see 2-CCGCAATT                1.80E-16 as potential motif for LBD family, https://jaspar.elixir.no/matrix/MA1673.2/ but you should prove this with p-value. I think you should provide. Why there are other motives

Fig. 6 is not understandable.

Remove all numbers from heatmap, mark all insignificant cell with white, and let shades of blue/red mean the positive/negative significance.

Provide link to group/trait definition (axes Y/X), this should be just lists 1,2,3…, add to the Figure caption ref. to Table S5 (it is good). I can not catch anything about the colors from Table S6. How groups are related to anything else? I can not find definitions somewhere in supplement or the manuscript.

If you propose artificial colors you should explain their meaning, see example

Filion, G. J., van Bemmel, J. G., Braunschweig, U., Talhout, W., Kind, J., Ward, L. D., Brugman, W., de Castro, I. J., Kerkhoven, R. M., Bussemaker, H. J., & van Steensel, B. (2010). Systematic protein location mapping reveals five principal chromatin types in Drosophila cells. Cell, 143(2), 212–224. https://doi.org/10.1016/j.cell.2010.09.009

Why three colors were selected for Table 7? What about other colors?

236

…Notably, three modules (yellow,black, and brown) exhibit strong correlations with the OE-PtrLBD39 trait data (P < 0.05, |R| > 0.85) (Figure 6),  (do you mean the strongest or strong?)

This is not clearly seen. Axis X notations 1…23 are in supplement, so explain there (caption or text) where the OE-PtrLBD39 trait data within these numbers.

Why correction for the multiple comparisons is absent? the number of cells is quite large in Figure 6. It is wrong that almost all cells either blue or red.

Comments on the Quality of English Language

Text is moderately good, but misprints are still present, e.g.

272

...Furthermore, this term encompasses genes associated with hormone signaling pathways, including auxin response factor ARFs and ERFs...

66

Therefore, we hypothesized that PtrLBD39’s ectopic expression disrupted.. ->

ectopic expression of PtrLBD39

I don't check all text, authors should do that. 
